# Integrin ligands block mechanical signal transduction in baroreceptors

Haiyan Zhao[1,2,3,*] , Ping Liu[1,3,*], Xu Zha[1], Sitao Zhang[4], Jiaqi Cao[1], Hua Wei[5], Meili Wang[1,3,6], Haixia Huang[1,3,6] , Wei Wang[1,3]

**Baroreceptors are nerve endings located in the adventitia of the carotid sinus and aortic arch. They act as a mechanoelectrical transducer that can sense the tension stimulation exerted on the blood vessel wall by the rise in blood pressure and transduce the mechanical force into discharge of the nerve endings. However, the molecular identity of mechanical signal transduction from the vessel wall to the baroreceptor is not clear. We discovered that exogenous integrin ligands, such as RGD, IKVAV, YIGSR, PHSRN, and KNEED, could restrain pressure-dependent discharge of the aortic nerve in a dose-dependent and reversible manner. Perfusion of RGD at the baroreceptor site in vivo can block the baroreceptor reflex. An immunohistochemistry study showed the binding of exogenous RGD to the nerve endings under the adventitia of the rat aortic arch, which may competitively block the binding of integrins to ligand motifs in extracellular matrix. These findings suggest that connection of integrins with extracellular matrix plays an important role in the mechanical coupling process between vessel walls and arterial baroreceptors.**

## Introduction

The baroreceptor reflex (depressor reflex) is one of the most important reflexes in maintaining the stability of arterial blood pressure (BP). The basic function of the baroreceptor reflex is to monitor arterial BP in real time. In detail, the BP signal is sensed by the arterial baroreceptor to regulate cardiovascular activity in a negative feedback manner through the integration of the medullary cardiovascular center to maintain the stability of BP. The function of the arterial baroreceptor is mainly to convert the mechanical stimulation applied on the blood vessel wall because of the change in BP to the nerve electrical signal that can be transmitted to the central nervous system, that is, mechanoelectrical transduction.

Mechanosensitive ion channels (MSCs) are considered to be an important molecular basis for this function. In recent years, studies have revealed that the epithelial sodium channel (ENaC), some members of the transient receptor potential family, some members of the acid-sensing ion channel family, PIEZO, and Tentonin 3 are involved in this process (Drummond et al, 2001; Lau et al, 2016; Zeng et al, 2018; Lu et al, 2020; Yan et al, 2021). However, there is no report on how the mechanical tension acting on the vascular wall is transmitted to MSCs on the nerve terminals of the baroreceptors. Under physiological conditions, the fluctuation in BP distorts the vascular wall and causes pressure-dependent discharge of the arterial baroreceptor, which indicates that there is a relatively stable link between the baroreceptor and the vascular wall. One study found that the baroreceptor enters the adventitia and separates into bundles, forming a complex sensory terminal region (Ghitani & Chesler, 2019; Min et al, 2019). Several layers of basal laminae are around the sensory terminals (Kimani, 1992). The basal laminae are composed of various ECM, which can guide nerve development (Rogers et al, 1983; Kaplan & Levenberg, 2022). The stress on the vascular wall may be transmitted to sensory terminals through these laminae. It is reasonable to speculate that there may be a relatively stable structural link between terminals of baroreceptors and ECM of the basal laminae, through which the stable mechanical coupling between the nerve terminals and the matrix can be maintained.

Only after mechanical signals acting on the baroreceptor are transmitted to MSCs can mechanoelectrical transduction be achieved. At present, two basic models are used to describe the gating of MSCs by mechanical force: the force-from-lipid model and the force-from-filament model. The force-from-lipid model holds that the mechanical force gating the channels comes from the lipid bilayer alone. The force-from-filament model favors the notion of channel gating via cytoskeletal or extracellular tethers that are directly or indirectly connected to the channels (Cox et al, 2019). Regardless, mechanical signals on the vascular wall need to be transmitted to baroreceptors. Although it has been reported that

---

[1]Department of Physiology and Pathophysiology, School of Basic Medical Sciences, Capital Medical University, Beijing, China    [2]Yanjing Medical College, Capital Medical University, Beijing, China    [3]Beijing Lab for Cardiovascular Precision Medicine, Beijing, China    [4]Department of Orthopedics, Xuanwu Hospital, Capital Medical University, Beijing, China    [5]Medical Experiment and Test Center, Capital Medical University, Beijing, China    [6]Beijing Key Laboratory of Metabolic Disorders Related Cardiovascular Diseases, Capital Medical University, Beijing, China

Correspondence: haixiah@ccmu.edu.cn
*Haiyan Zhao and Ping Liu contributed equally to this work

ENaC is in direct contact with ECM through two glycosylated asparagine residues (Barth et al, 2021), there is no other evidence that MSCs on baroreceptors can interact directly with ECM. It is necessary to explore how stretch stimulation of the vascular wall is transmitted to baroreceptors.

Cell–ECM interactions are mediated largely by integrins, a class of heterodimeric cell adhesion molecules. The binding of integrins and ligand motifs in ECM forms the mechanical coupling between cells and their surroundings. It has been reported that integrins may be associated with the activation of MSCs in the process of mechanical signal sensing and transmission (Arcangeli & Becchetti, 2006). Shakibaei's study showed that $\beta_1$-integrins colocalize with Na, K-ATPase, ENaC, and voltage-activated calcium channels in mechanoreceptor complexes of mouse limb-bud chondrocytes and the influx of calcium or sodium through putative MSCs may be regulated more effectively if such channels are organized around integrins (Shakibaei & Mobasheri, 2003). MSCs can be activated when mechanical force is transferred from ECM to integrins (Potla et al, 2020). Integrin-mediated signal transduction, including mechanical signal transduction, is achieved by the specific combination of integrins and specific ligands in ECM. Arg-Gly-Glu (RGD) is the most important and common ligand of integrins existing on a variety of ECM proteins (Jalali et al, 2001; Ludwig et al, 2021). In addition to RGD, Ile-Lys-Val-Ala-Val (IKVAV) and Tyr-Ile-Gly-Ser-Arg (YIGSR), which are specifically located in laminin, and Pro-His-Ser-Arg-Asn (PHSRN) and Lys-Asn-Glu-Glu-Asp (KNEED), which are specifically located in fibronectin, can also bind to integrins to complete signal transduction (Wong et al, 2002; Wu et al, 2018; Jain & Roy, 2020). Whether the binding of integrins to ECM plays a similar role in mechanical signal transduction in the arterial baroreceptor is worth exploring.

In this study, the structural connection between integrin and ECM was blocked by intravascular perfusion of exogenous free ligands, such as RGD and IKVAV. The effects on both the discharge of arterial baroreceptor and baroreceptor reflex were investigated to explore the role of integrins binding to ECM in mechanoelectrical transduction of arterial baroreceptors. In addition, the arterial baroreceptors of mice and the structures of their blood vessels are too small, so it is difficult to conduct vascular perfusion studies in vitro or in vivo. Like other researchers, we used rabbits with large blood vessels to conduct perfusion experiments (Liu et al, 2021).

## Results

### RGDF inhibited pressure-dependent discharges of the perfused baroreceptors in vitro

When the CS-SN was perfused with normal perfusate, the carotid baroreceptor was characteristically discharged in accordance with the pressure surge (60/100 mmHg) in the carotid sinus (Fig 1A and B). Switching from the normal perfusate to the perfusate containing incremental concentrations of RGDF (1, 2, and 4 mM in turn) for 5 min, the activity of the baroreceptor was inhibited by RGDF in a concentration-dependent manner (Fig 1A). The discharge frequency of the sinus nerve was reduced to 62.21% ± 1.67% ($P$ < 0.0001, n = 5),

30.93% ± 5.59% ($P$ < 0.0001, n = 5), and 2.44% ± 1.04% ($P$ < 0.0001, n = 5) (Fig 1D), respectively. In addition, after washing with normal perfusion solution for 10 min, the pressure-dependent discharge of the baroreceptor could recover to 85.78% ± 1.88% of the control (Fig 1A and E). However, perfusates containing the control peptide RGEF (as the control peptide of RGDF) at the same concentration had no effect on the discharge (Fig 1C and F), suggesting that the blocking action was specific for RGDF.

To study the effect of RGD on the sinus nerve discharge–sinus pressure relationship, we recorded sinus nerve discharge in step fluctuating pressure stimulation mode before and after the application of RGD. The perfusing pressure function of $A·sin(2\pi ft)+B$ was used, where $A$ was pulse pressure, which was set at 40 mmHg, $f$ was frequency set at 1 Hz, and $B$ was the mean pressure, which was 20, 40, 60, 80, 100, 120, 140, or 180 mmHg, changing in jumps. First, the discharge of the sinus nerve was recorded under step fluctuating pressure stimulation with normal perfusion (Fig 2A). Then, after perfusion with perfusate containing 4 mM RGDF under a pressure surge (80/120 mmHg) for 5 min, sinus nerve discharge was recorded under the same step fluctuating pressure stimulation (Fig 2B). The results showed that the carotid sinus pressure–dependent discharge disappeared, even at the higher pressure levels (Fig 2B and D). And sinus nerve discharge can be recovered after washing with ordinary perfusate (Fig 2C).

### RGDF blocks the baroreceptor reflex in vivo

To determine whether RGDF can block the baroreceptor reflex in vivo, we used an isolated perfused carotid sinus that was separated from the system with an intact sinus nerve. In this model, the baroreceptor reflex changed from a closed-loop regulation to an open-loop regulation. A typical baroreceptor reflex was confirmed. That is, when the perfusing pressure jumped from 150/200 to 0/50 mmHg, increases in the mean system BP by 17.72 ± 1.73 mmHg ($P$ = 0.0019, n = 3) and the heart rate (HR) by 4.541 ± 1.445 beats/min ($P$ = 0.0265, n = 3; Fig 3A, C, and D) were confirmed. In addition, the stepped intracarotid sinus pressure (ISP) from 0/50 to 150/200 mmHg resulted in decreases in both the mean system BP by 23.40 ± 5.181 mmHg ($P$ = 0.0087, n = 3; Fig 3B and E) and the HR by 5.863 ± 1.077 beats/min in 7 s ($P$ = 0.0066, n = 3; Fig 3B and F). The BP and HR did not change with the same stepped ISP stimulation protocol after perfusing the isolated carotid sinus with perfusate containing 4 mM of RGDF (Fig 3A–F). However, the mean arterial BP increased by 18.21 ± 4.101% mmHg ($P$ < 0.0001, n = 3) at 150/200 mmHg fluctuating ISP after perfusing the isolated carotid sinus with RGDF (Fig 3G). The function curve of the baroreceptor reflex changed from an S-shaped curve to a straight line (Fig 3H). This result indicated that RGDF inhibits baroreceptor reflex activity by blocking baroreceptors.

### Blocking effects of other integrin ligands on sinus nerve discharge

Other integrin ligands, such as PHSRN, KNEED, IKVAV, and YIGSR, were used to perfuse the isolated CS-SN specimen. The results showed that PHSRN dose-dependently blocked the pressure-dependent discharge of the sinus nerves and that the discharge

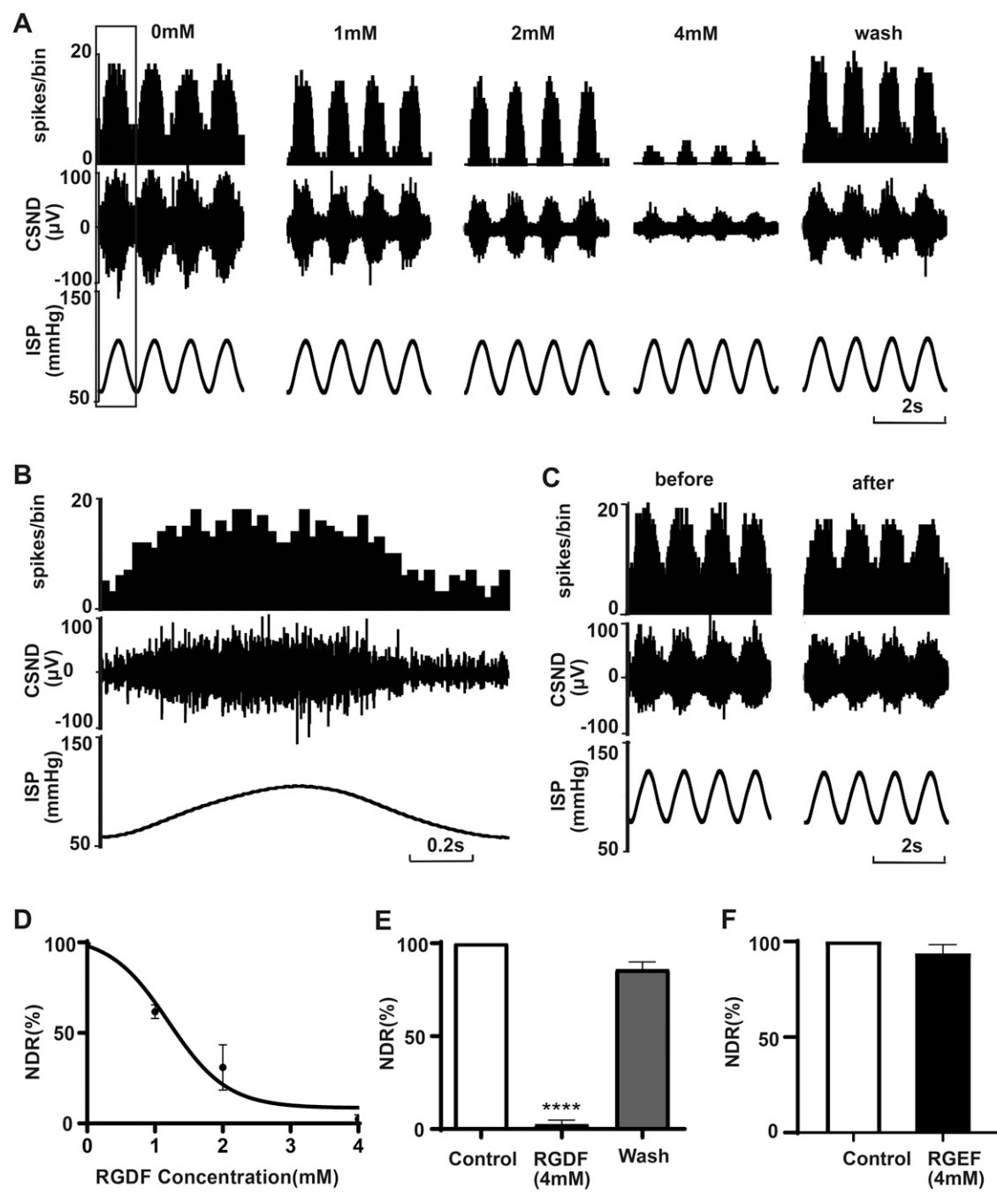

**Figure 1. Specific inhibitory effect of RGDF on baroreceptor activity.**
An original record showing the effect of RGDF on the activity of the sinus nerve. From the bottom to the top are intracarotid sinus pressure, synchronously recorded carotid sinus nerve discharge, and the frequency histogram of the sinus nerve discharge (bin value is 0.01 s). **(A)** Discharges of the sinus nerve after perfusion with perfusate containing 0, 1, 2, and 4 mM RGDF, respectively, for 5 min and the recovery of the discharge of the sinus nerve after washing with normal perfusion solution for 10 min. **(A, B)** Expansion diagram in the block is shown in (A). **(C)** Discharge of the sinus nerve before and after perfusion with perfusate containing 4 mM RGEF. **(D)** Dose–effect curve of the inhibitory effect of RGDF on sinus nerve discharge (n = 5). The longitudinal coordinate is the normalized discharge rate (the number of discharges in one intracarotid sinus pressure cycle standardized by the number of discharges in the control group). **(E, F)** Statistical results of the effect of RGDF ((E), n = 5) and its control peptide RGEF ((F), n = 3).

was almost completely blocked at 4 mM of PHSRN, which could be recovered after washing (Fig 4A). Other ligands also dose-dependently blocked the pressure-dependent discharge of the sinus nerves (Fig 4B). PHSRN (4 mm), KNEED (4 mm), IKVAV (4 mm), and YIGSR (4 mm) blocked the pressure-dependent discharge of the sinus nerves to 1.993% ± 0.3783%, 2.08% ± 1.083%, 2.068% ± 0.573%, and 2.364% ± 0.6525% of the original discharge, respectively (Fig 4C).

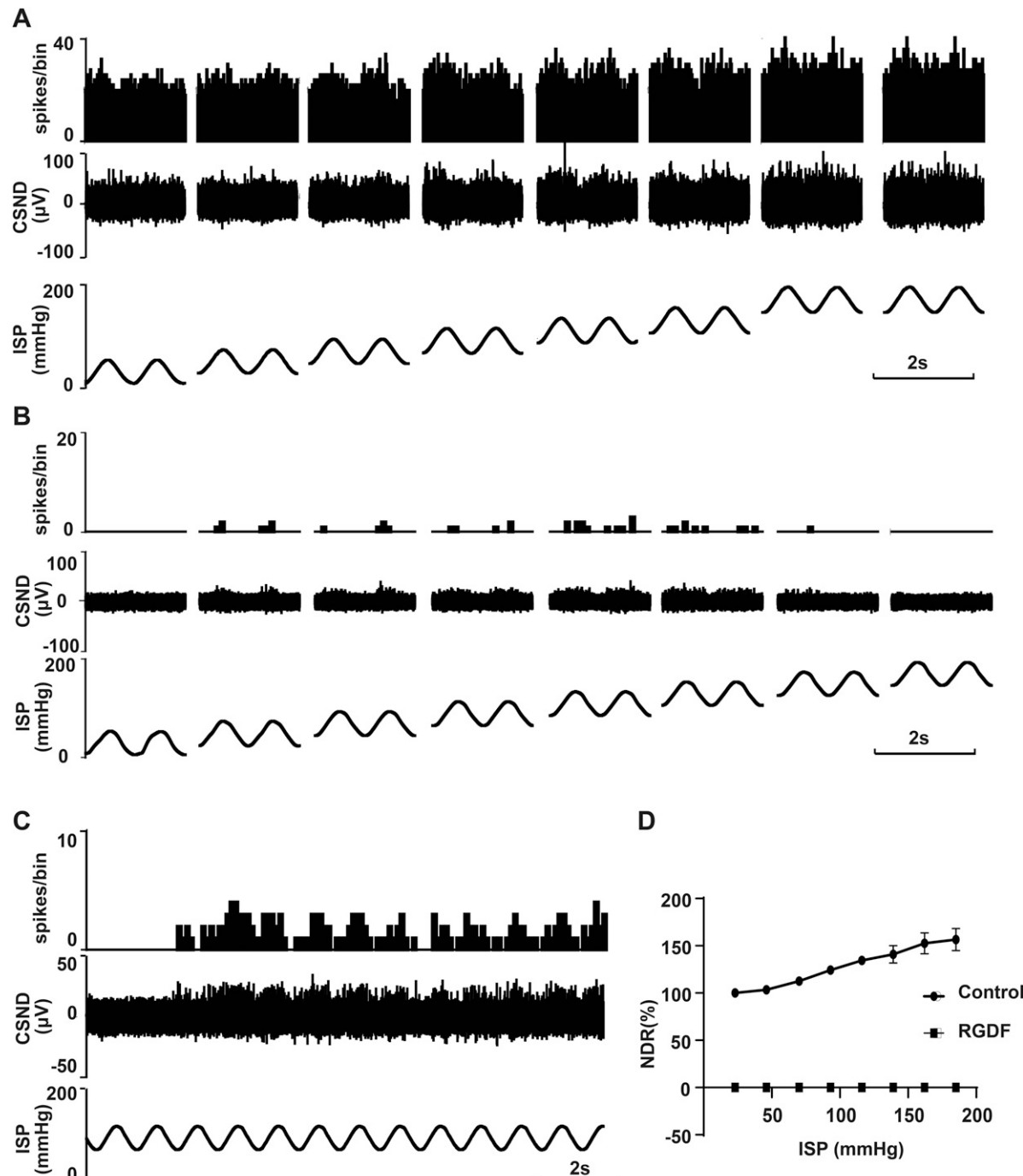

**Figure 2.  Effect of RGDF on neural discharge under different pressure stimulations.**
**(A, B)** Original records showing sinus nerve discharges under the step fluctuating pressure stimulation of ISP before ((A), control group) and after perfusion with perfusate containing 4 mm of RGDF. **(B)** From the bottom to the top are the ISP, CSND, and frequency histogram of the sinus nerve discharge (bin value is 0.01 s). **(C)** Process of gradual recovery of sinus nerve discharge after washing with ordinary perfusate. **(D)** Relationship between sinus pressure and sinus nerve discharge before and after perfusion with 4 mm of RGDF. The longitudinal coordinate is the NDR' (the discharge number in each ISP cycle under different pressures standardized by the discharge number in the basic state in the control group. We named it NDR' to distinguish it from the previous NDR).

## Binding of RGDF to aortic baroreceptors

To determine whether the inhibitory effect of exogenous RGDF on sinus nerve discharge is due to the binding of RGDF to integrins that competitively inhibited the binding of endogenous ligands in ECM to integrins, the binding of RGDF to aortic baroreceptors was studied. After perfusing the aortic arch specimen in vitro with perfusate containing rhodamine B–conjugated RGDF (2 mM) at

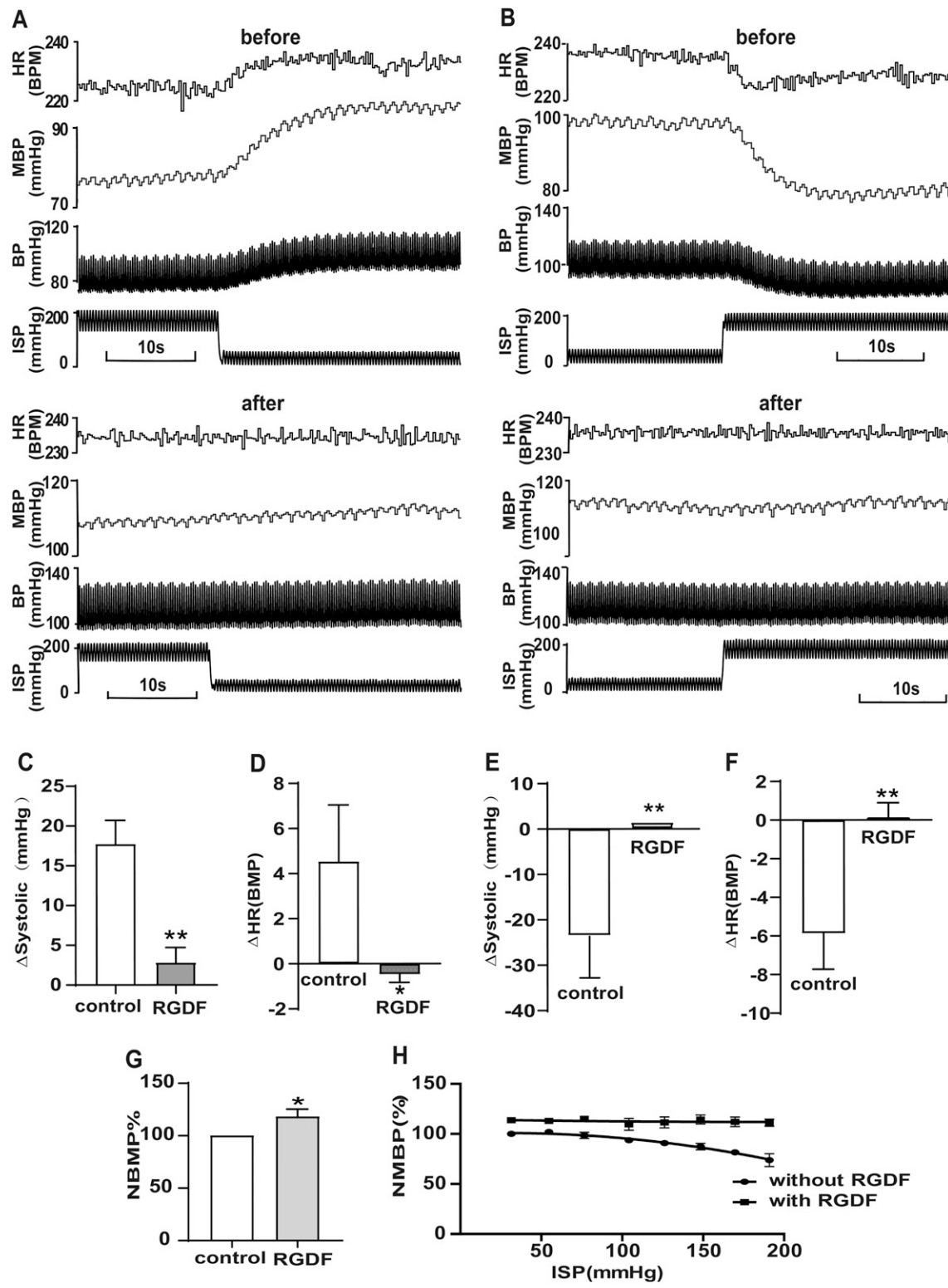

**Figure 3. Inhibitory effect of RGDF on baroreceptor reflex activity.**
**(A, B)** Original records of systemic arterial blood pressure under the control of ISP. From the bottom to the top are the ISP in controlling, systemic arterial blood pressure (BP), mean arterial blood pressure (MBP), and heart rate (HR) calculated from the original record of systemic arterial blood pressure. **(A)** Systemic arterial HR, MBP, and HR changes when the ISP was reduced from 150/200 to 0/50 mmHg before and after perfusion of the isolated carotid sinus with 4 mm of RGDF. **(B)** Systemic arterial BP, MBP, and HR changes when the ISP was increased from 0/50 to 150/200 mmHg, before and after perfusion of the isolated carotid sinus with 4 mm of RGDF. **(C, D)** Statistical graph of systolic BP change (C) and HR change (D) (n = 3) when the ISP was reduced from 150/200 to 0/50 mmHg before and after perfusion with RGDF. **(E, F)** Statistical

80/120 mmHg for 10 min, the epicardial membrane containing baroreceptors was stripped off the aortic arch and stained with the nerve marker PGP9.5. We found that exogenous RGDF–rhodamine B was located on the nerve endings under the epicardium of the aortic arch (Fig 5). This binding may interrupt the mechanical coupling between the vascular wall and the arterial baroreceptor.

# Discussion

The stable mechanical coupling between the baroreceptor and the vascular wall is the basis to complete mechanoelectrical transduction. A relatively stable structural link between the vascular wall and the baroreceptor is necessary. Through this link, nerve endings of baroreceptors are fixed in ECM of the vascular wall. However, it is currently unknown what the molecular basis of this stable mechanical coupling is.

Integrins serve as crucial sites for both outside-in and inside-out mechanotransduction, which has been confirmed in many processes. Integrin-mediated adhesion has a central role in cellular mechanosensing: it is the physical link between individual cells and their surrounding ECM (Schwartz, 2010). Thus, cell-matrix adhesions can be considered mechanical connectors. On the intracellular side, they are linked to the cell cytoskeleton, and on the extracellular side, they are coupled to extracellular scaffolds formed by proteins, such as fibronectin and laminin, which contain specific attachment sites for integrin receptors (Zamir & Geiger, 2001).

It has been found that many integrins are expressed in the nervous system. Integrins play an important role in the growth of axons and dendrites on ECM components (Reichardt & Tomaselli, 1991; Tomaselli et al, 1993). Interactions of cell surface $\alpha_3/\beta_1$ and $\alpha_6/\beta_1$ integrins with laminin-1 likely mediate the growth of avian ciliary ganglion neurons during pathfinding in vivo (Weaver et al, 1995). In addition, multiple errors in axon pathfinding were identified in Drosophila null mutants lacking integrin $\alpha$ subunits ($\alpha$PS1 and $\alpha$PS2). Loss of RGD-dependent PS2 integrin ($\alpha$PS2 null mutation) leads to higher axon guidance errors than does loss of the laminin-binding PS1 integrin ($\alpha$PS1 null mutation) (Hoang & Chiba, 1998).

In this study, we found that perfusing the CS-SN specimen with peptides containing RGD, an important integrin ligand, can inhibit the discharge of sinus nerve in a concentration-dependent manner, and the blocking effect was reversible. These results suggest that the mechanical coupling between the vascular wall and baroreceptor may be realized by the binding of integrins to specific sites in ECM, and mechanical coupling is a dynamic binding. The results of the morphological study further confirmed that exogenous RGD could bind to the baroreceptor site, suggesting that exogenous RGD may block the physical coupling between the vascular wall and nerve endings by occupying the ECM protein binding site of integrins on the baroreceptors. Many other sequences in ECM, such as IKVAV and YIGSR, which are specifically located on laminin, and

PHSRN and KNEED, which are specifically located on fibronectin, are integrin binding sites too. We found that exogenous KVAV, YIGSR, PHSRN, and KNEED peptides could also inhibit sinus nerve discharge in addition to the RGD peptide. Therefore, we speculate that the mechanical signal on the vascular wall is transmitted to the nerve endings by the combination of various specific short peptides in ECM and integrins on the baroreceptor (Fig 6).

The transmission of the mechanical signal acting on the vascular wall to MSCs on the baroreceptor is the key link in the process of mechanoelectrical transduction. $\alpha_2\beta_1$ integrin antibody can reduce the nerve reaction caused by pressing the skin (Matthews et al, 2006). The direct force application to cell surface $\beta_1$ integrins using magnetic tweezers results in rapid calcium influx in bovine capillary endothelial cells, and this response can be blocked by general MSC inhibitor gadolinium chloride (Thodeti et al, 2009). Forces applied on $\beta_1$ integrins result in ultra-rapid (within 4 ms) activation of calcium influx through TRPV4 channels (Matthews et al, 2010). These reports show that integrins are involved in the transmission of extracellular mechanical signals to MSCs on the cell membrane. It is reasonable to speculate that the mechanical signal on the vascular wall is transmitted to integrins through ECM, and then transmitted to MSCs, which activates MSCs to produce receptor potential (Fig 6). The ion channels on baroreceptors form a compact complex and function synergistically in the mechanoelectrical transduction of arterial baroreceptors (Yan et al, 2021). The role of integrins in transmitting mechanical signals to MSCs and the relative mechanisms deserve attention.

Integrin ligands dose-dependently block the baroreceptor function both in vitro and in vivo, which suggests connection of integrins with ECM is indispensable for mechanic signal transduction in baroreceptors. It will be necessary and valuable to use gene knockout technology to further prove this role of integrin in the future.

Organisms respond to various mechanical stimuli through various mechanoreceptors. Similar to the baroreceptor, many mechanoreceptors are nerve endings in the sensory area, such as the Pacinian corpuscle, Meissner corpuscle, and muscle spindle. These mechanoreceptors have a common feature in structure. That is, the terminals of these receptors are surrounded by matrix proteins, and some even form special structures. Integrins may play an important role in mediating the transmission of extracellular mechanical signals to mechanoreceptors. The current study provides a new clue to uncover the mechanical signal transduction mechanism of mechanoreceptors.

# Materials and Methods

### Rabbit and mice

Animal care, ethical use, and protocols were approved by the Institutional Animal Care and Use Committee of the Capital Medical University, Beijing, China.

graph of systolic BP change (E) and HR change (F) (n = 3) when the ISP was increased from 0/50 to 150/200 mmHg before and after perfusion with perfusate containing 4 mM of RGDF. **(G)** Effects of RGDF on the mean arterial blood pressure when ISP was controlled at the base step. The longitudinal coordinate is the normalized mean arterial blood pressure (NMBP, the mean arterial blood pressure compared with that in the control group). **(H)** Baroreceptor reflex function curve moves up significantly after perfusion of the isolated carotid sinus with perfusate containing 4 mM of RGDF. The longitudinal coordinate is the normalized mean arterial blood pressure (NMBP, the mean arterial blood pressure at each ISP compared with that when ISP was controlled at the base step).

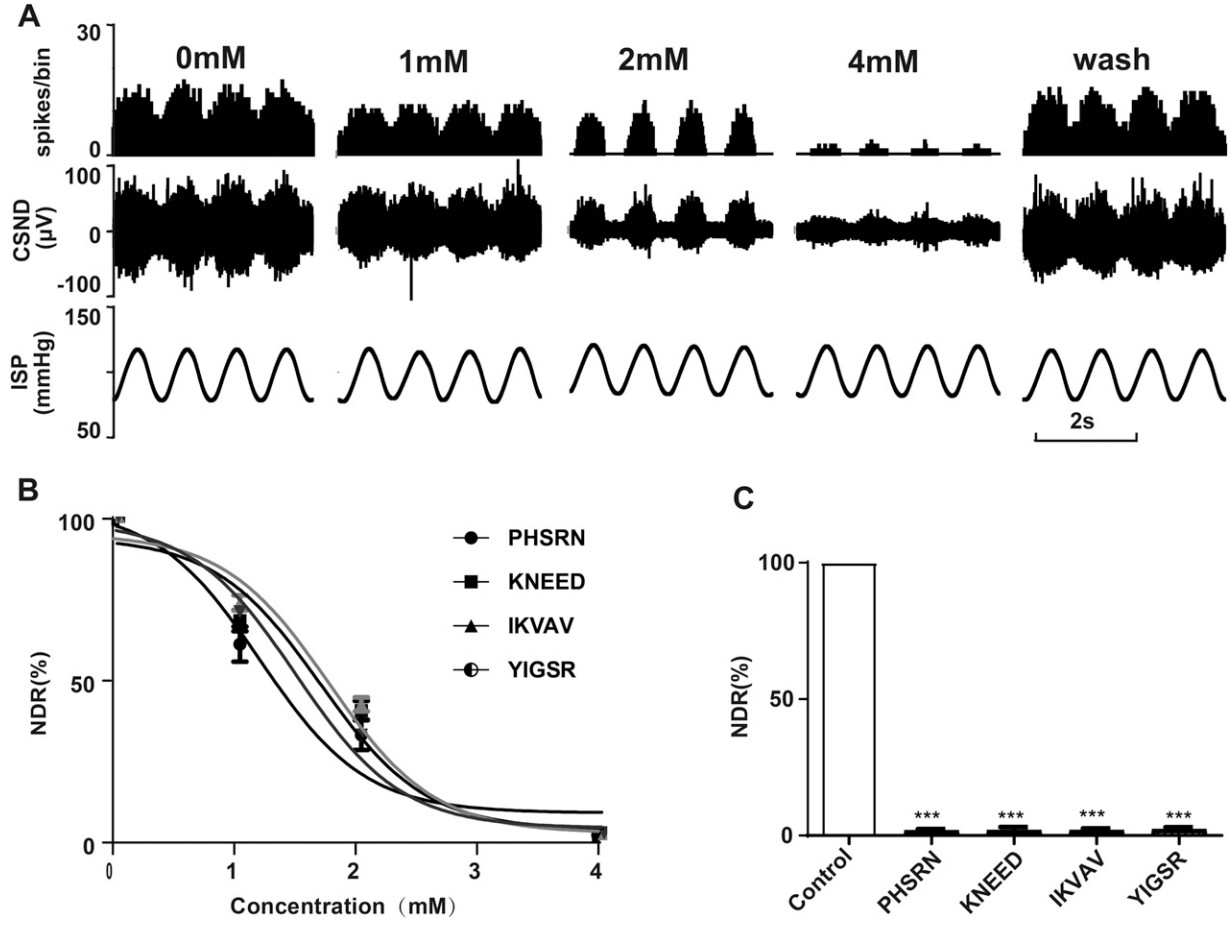

**Figure 4. Inhibitory effect of other integrin ligands on baroreceptor activity.**
**(A)** Original records showing the effect of PHSRN on the activity of the sinus nerve. From bottom to top are the ISP, synchronously recorded sinus nerve discharge, and the frequency histogram of the sinus nerve discharge (bin value is 0.01 s). **(A)** Discharges of the sinus nerve after perfusion with perfusate containing 0, 1, 2, and 4 mM of PHSRN, respectively, and the recovery of sinus nerve discharge after washing with normal perfusion solution. **(B)** Dose–effect curve of PHSRN, KNEED, IKVAV, and YIGSR on the inhibition of sinus nerve discharge (n = 3). The longitudinal coordinate is the NDR. **(C)** Statistical results of the effect of 4 mM of PHSRN (n = 3), 4 mM of KNEED (n = 3), 4 mM of IKVAV (n = 3), and 4 mM of YIGSR (n = 3).

## Recording discharges of arterial baroreceptors in vitro

The experiment was performed as described previously (Seagard et al, 1993). In brief, adult rabbits (2 ± 0.3 kg, male or female) were intravenously anesthetized with urethane (1 g/kg) and then intubated endotracheally. Under spontaneous breathing, the carotid sinus–sinus nerve specimen (CS-SN) containing the carotid artery (7 mm long), external carotid artery (5 mm long), internal carotid artery (5 mm long), and sinus nerve (7 mm long) was isolated. Then, the preparation was mounted into a custom-made perfusing chamber filled with flowing superfusion solution (5 ml/min). The internal carotid artery was ligated beyond the sinus, and the common carotid artery and external carotid artery were cannulated as the perfusing inlet and outlet, respectively. The superfusion solution was the same as the normal perfusate containing (in mmol/l) NaCl, 140; KCl, 6; MgSO$_4$, 1.2; glucose, 5.5; CaCl$_2$, 1.1; and Hepes, 10. The pH was adjusted with Hepes–NaOH to 7.38–7.42 at 37°C, and the solution was bubbled with 100% oxygen to inhibit the chemoreceptors (Seagard et al, 1993). The preparation was perfused at a rate of 5 ml/min with the normal perfusate alone or with the perfusate supplemented with one of the following peptides: RGDF (RGD) (04010046165; ChinaPeptides), RGEF (RGE) (04010024270; ChinaPeptides), IKVAV (04010019354; ChinaPeptides), YGSA (04010068582; ChinaPeptides), PHSRN (04010024270; ChinaPeptides), or KNEED (04010065581; ChinaPeptides).

As the reference electrode, a stainless-steel wire (0.1 mm in diameter) with 1 mm of length exposed at one end was placed under the carotid sinus and immersed in the superfusion solution. Another stainless-steel wire (0.1 mm in diameter, 5 mm in length), which was immersed in paraffin oil that was in a separate chamber next to the preparation chamber, was used as the recording electrode. Afferent activities were amplified by a factor of 5,000, filtered at a passband of 300 Hz to 10 kHz, and digitized at 20 kHz using a data acquisition system (AI 402, CyberAmp 380, pCLAMP 10.0, and Digidata 1440; Molecular Devices). Baroreceptor discharges were analyzed offline mainly using Spike2 (Cambridge Electronic Design Limited). Because the recorded discharge frequency of each specimen is quite different, we have standardized the recorded discharge frequency of sinus nerve for the convenience of data analysis. The number of

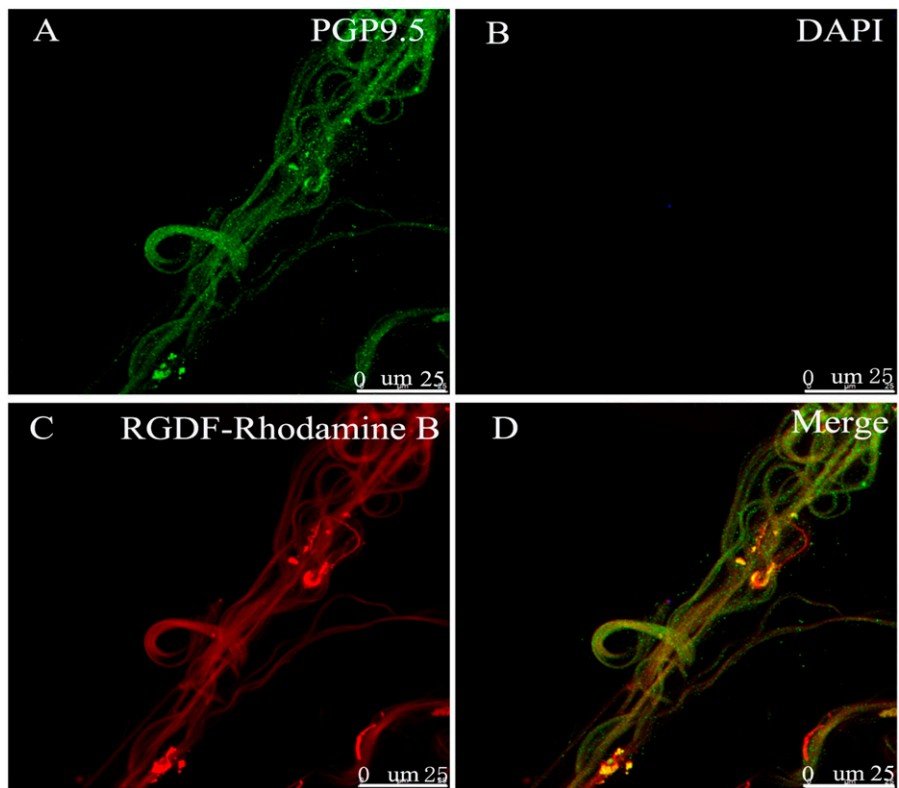

**Figure 5. Combination of RGDF and arterial baroreceptor.**
**(A)** PGP9.5 (green) immunoreactive nerve endings located in the adventitia of the aortic arch. **(B)** Nuclei are labeled with DAPI (blue) in (B). **(C)** Nerve endings combined with exogenous RGDF–rhodamine B (red). **(A, B, C, D)** Merged image of Panels (A, B, C).

discharges in one ISP cycle standardized by the number of discharges in the control group was named the normalized discharge rate (NDR,

$$NDR = \frac{\text{number of discharges in one ISP cycle}}{\text{number of discharges in one ISP cycle in the control group}} \times 100\%).$$

### Perfusing carotid sinus and recording arterial BP in vivo

Adult rabbits were anesthetized and intubated endotracheally as above. Then, an isolated perfusion model of the carotid sinus in vivo was set up on either side of the neck. The main procedures included the following: (1) ligating the internal carotid artery downstream of the carotid sinus; (2) for a perfusing inlet, intubating the common carotid artery toward the sinus and ligating this artery at the upstream side; (3) for a perfusing outlet, intubating the external carotid artery toward the sinus and ligating this artery at the cranial side; and (4) placing a pressure transducer (MLT844; ADInstruments) at the perfusing outlet for recording the ISP. The same perfusate was used as in the experiments in vitro. In addition, the system pressure was recorded by intubating the femoral artery with another pressure transducer of the same type. The sinus pressure and system pressure were sampled by a data acquisition system (PowerLab/ LabChart 5; ADInstruments) at 1 kHz. The HR was calculated off-line based on the waves of system pressure.

### Controlling the ISP in vitro and in vivo

The same method was used to perfuse the carotid sinus in vitro and in vivo. To simulate the physiological stimulation to the

baroreceptor, the perfusing pressure with sine waves was driven by a custom-made pressure servo system (PRE-U; Hoerbiger). The perfusing pressure function of $A \cdot sin(2\pi ft)+B$ was used, where $A$ is the amplitude, $f$ is frequency, $t$ is time, and $B$ is the mean pressure. For simplicity, the perfusing pressure was normally expressed as trough/ peak in mmHg as the system pressure. ISP was recorded with a transducer (YH-4; Beijing Institute of Aerospace Medical Engineering or MLT844; ADInstruments), which was placed adjacent to the carotid sinus at the perfusate outlet and sampled synchronously with the activity of the sinus nerve in vitro or with the system pressure in vivo.

### RGD-binding experiment on the aortic arch

The aortic arch (8 mm long) was obtained from the SD rat (200–300 g) under intraperitoneal anesthesia with 20% urethane (0.5 ml/100 g) and positive pressure ventilation. The specimen was perfused with perfusate containing 2 mM rhodamine B–conjugated RGDF. The intravascular pressure was controlled at 80–120 mmHg. After 10 min, the specimens were placed in 4% paraformaldehyde at 4°C overnight. Then, they were used for subsequent immunohistochemistry experiments.

### Immunohistochemical study of the aortic baroreceptor

After the RGD-binding experiment, the endothelium and smooth muscle layer were peeled off the fixed aortic arch with fine tweezers under a stereomicroscope. The specimen was washed and permeabilized with 1% Triton X in PBS. Then, Triton X was washed away with PBST (0.3% Triton X in PBS). After blocking with 3% bovine serum albumin in PBST at 4°C overnight, the specimen was incubated in PBST containing rabbit anti-mouse PGP9.5 polyclonal

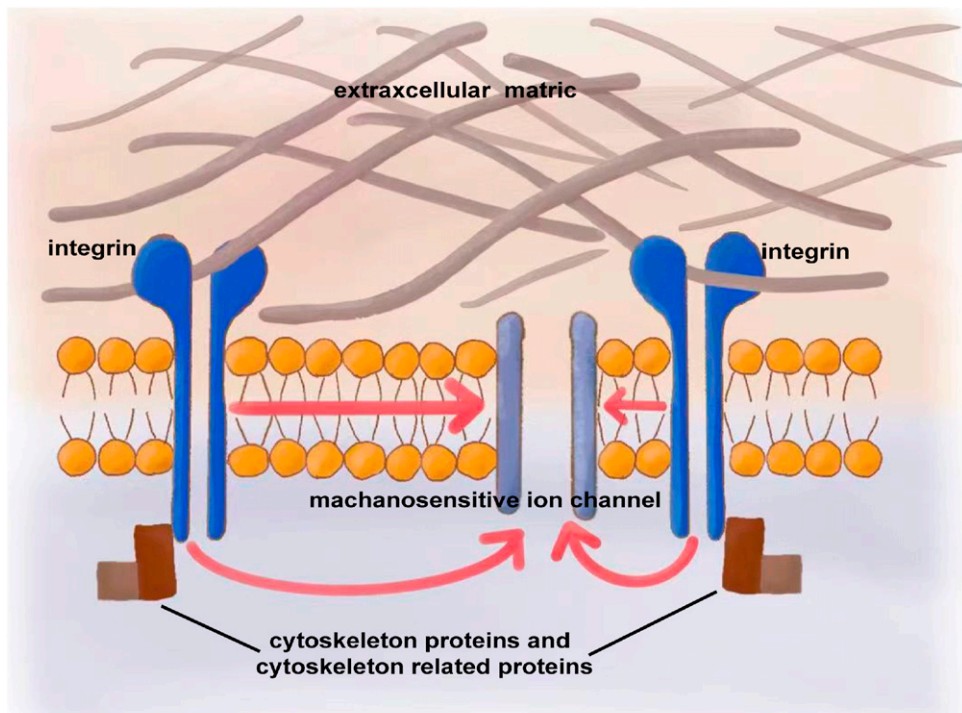

**Figure 6. Model diagram of the integrin-based mechanical signal transmission mechanism of arterial baroreceptors.** Integrins located on the nerve endings of arterial baroreceptors recognize and bind to ligand amino acid sequence on ECM proteins in the vascular wall to transmit mechanical stimulation on the vascular wall to the nerve endings. Mechanical stimulation is further transmitted through membrane lipids or cytoskeletal proteins and cytoskeletal-related proteins to mechanosensitive ion channels.

antibodies (1:100, ab108986; Abcam) at 4°C overnight, washed with PBST, and incubated in PBST containing Alexa Fluor 488–conjugated goat anti-rabbit secondary antibodies (1:200, ab150077; Abcam) for 3 h at 4°C. Nuclei were stained with 4',6-diamino-2-phenylindole hydrochloride (DAPI, 10236276001; Roche) at 4°C for 30 min. The specimen was mounted with 90% glycerol on thin coverslips. Slides were visualized using the Leica LAS AF-TCS SP5 confocal microsystem (Leica). Rhodamine B was excited at 540 nm and detected at 625 nm. Alexa Fluor 488 was excited at 488 nm and detected at 494–517 nm. DAPI was excited at 405 nm and detected at 420–480 nm.

## Statistical analysis

The data were processed by Prism 5.0 statistics software and compared by repeated measurement analysis of variance followed by Dunnett's test. All statistical data are expressed as the mean ± SE. The statistical significance was set at $P < 0.05$.

# Supplementary Information

# Acknowledgements

This work was supported by grants from the Nature and Science Foundation of China (31871147 and 32071130), Beijing Natural Science Foundation Program and Scientific Research Key Program of Beijing Municipal Commission of Education (KZ202010025038), and Scientific Research Foundation of Yanjing Medical College of Capital Medical University (20kyqd03).

## Author Contributions

H Zhao: data curation, investigation, project administration, and writing—original draft.
P Liu: resources, supervision, investigation, and methodology.
X Zha: investigation and methodology.
S Zhang: data curation and investigation.
J Cao: data curation and investigation.
H Wei: resources and writing—original draft.
M Wang: methodology and writing—original draft.
H Huang: conceptualization, resources, supervision, funding acquisition, investigation, methodology, project administration, and writing—review and editing.
W Wang: supervision, funding acquisition, methodology, project administration, and writing—review and editing.

## Conflict of Interest Statement

The authors declare that they have no conflict of interest.

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
