## [Reviewer comments · Life Science Alliance]

Life Science Alliance

Integrin ligands block mechanical-signal transduction in baroreceptors

Haiyan Zhao, Ping Liu, Xu Zha, Sitao Zhang, Jiaqi Cao, Hua Wei, Meili Wang, Haixia Huang, and Wei Wang
DOI: <https://doi.org/10.26508/lsa.202201785>

Corresponding author(s): Haixia Huang, Capital Medical University

Review Timeline:	Submission Date:	2022-10-25
	Editorial Decision:	2022-10-26
	Revision Received:	2022-11-25
	Editorial Decision:	2022-11-28
	Revision Received:	2022-12-02
	Accepted:	2022-12-08

Transaction Report:

Please note that the manuscript was previously reviewed at another journal and the reports were taken into account in the decision-making process at Life Science Alliance. Since the original reviews are not subject to Life Science Alliance's transparent review process policy, the reports and author response cannot be published.

October 26, 2022

Re: Life Science Alliance manuscript #LSA-2022-01785-T

Prof. Haixia Huang
Capital Medical University
Department of Physiology and Pathophysiology
You An Men Wai Street Xitoutiao10
Beijing 100069
China

Dear Dr. Huang,

Thank you for submitting your manuscript entitled "Connection of integrins with ECM is indispensable for mechanical-signal transduction in baroreceptors" to Life Science Alliance. We invite you to re-submit the manuscript, revised to address the Reviewers' concerns.

Thank you for this interesting contribution to Life Science Alliance. We are looking forward to receiving your revised manuscript.

Sincerely,

B. MANUSCRIPT ORGANIZATION AND FORMATTING:

November 28, 2022

RE: Life Science Alliance Manuscript #LSA-2022-01785-TR

Prof. Haixia Huang
Capital Medical University
Department of Physiology and Pathophysiology
You An Men Wai Street Xitoutiao10
Beijing 100069
China

Dear Dr. Huang,

Thank you for submitting your revised manuscript entitled "Integrin ligands block mechanical-signal transduction in baroreceptors". We would be happy to publish your paper in Life Science Alliance pending final revisions necessary to meet our formatting guidelines.

- please add the Twitter handle of your host institute/organization as well as your own or/and one of the authors in our system
- please consult our manuscript preparation guidelines <https://www.life-science-alliance.org/manuscript-prep> and make sure your manuscript sections are in the correct order
- please use the [10 author names, et al.] format in your references (i.e. limit the author names to the first 10)
- please add a callout for Figure 1B and Figure 2C to your main manuscript text
- please add an Ethics Statement to your Materials and Methods section to indicate that approval was granted for the animal work, and who provided that approval

Figure Check:

- you may upload Figure 6 as a Graphical Abstract instead, if you prefer

A. FINAL FILES:

B. MANUSCRIPT ORGANIZATION AND FORMATTING:

Sincerely,

December 8, 2022

RE: Life Science Alliance Manuscript #LSA-2022-01785-TRR

Prof. Haixia Huang
Capital Medical University
Department of Physiology and Pathophysiology
You An Men Wai Street Xitoutiao10
Beijing 100069
China

Dear Dr. Huang,

Thank you for submitting your Research Article entitled "Integrin ligands block mechanical-signal transduction in baroreceptors". It is a pleasure to let you know that your manuscript is now accepted for publication in Life Science Alliance. Congratulations on this interesting work.

DISTRIBUTION OF MATERIALS:

Again, congratulations on a very nice paper. I hope you found the review process to be constructive and are pleased with how the manuscript was handled editorially. We look forward to future exciting submissions from your lab.

Sincerely,
